# Identification of HPV16 Lineages in South African and Mozambican Women with Normal and Abnormal Cervical Cytology

**DOI:** 10.3390/v16081314

**Published:** 2024-08-18

**Authors:** Cremildo Maueia, Olivia Carulei, Alltalents T. Murahwa, Ongeziwe Taku, Alice Manjate, Tufária Mussá, Anna-Lise Williamson

**Affiliations:** 1Division of Medical Virology, Department of Pathology, Faculty of Health Sciences, University of Cape Town, Cape Town 7925, South Africa; olivia.carulei@uct.ac.za (O.C.); alltalents.murahwa@uct.ac.za (A.T.M.); takuongeziwe@gmail.com (O.T.); anna-lise.williamson@uct.ac.za (A.-L.W.); 2Departamento de Microbiologia, Faculdade de Medicina, Universidade Eduardo Mondlane, Maputo P.O.Box 257, Mozambique; alimanjate28@gmail.com (A.M.); tufariamussa@gmail.com (T.M.); 3Instituto Nacional de Saúde, Maputo 3943, Mozambique; 4Institute of Infectious Disease and Molecular Medicine, University of Cape Town, Cape Town 7925, South Africa; 5SAMRC Gynaecological Cancer Research Centre, Faculty of Health Sciences, University of Cape Town, Cape Town 7925, South Africa

**Keywords:** HPV16, lineages, phylogeny, cytology

## Abstract

Background: Human papillomavirus 16 (HPV16) is an oncogenic virus responsible for the majority of invasive cervical cancer cases worldwide. Due to genetic modifications, some variants are more oncogenic than others. We analysed the HPV16 phylogeny in HPV16-positive cervical Desoxyribonucleic Acid (DNA) samples collected from South African and Mozambican women to detect the circulating lineages. Methods: Polymerase chain reaction (PCR) amplification of the long control region (LCR) and 300 nucleotides of the E6 region was performed using HPV16-specific primers on HPV16-positive cervical samples collected in women from South Africa and Mozambique. HPV16 sequences were obtained through Next Generation Sequencing (NGS) methods. Geneious prime and MEGA 11 software were used to align the sequences to 16 HPV16 reference sequences, gathering the A, B, C, and D lineages and generating the phylogenetic tree. Single nucleotide polymorphisms (SNPs) in the LCR and E6 regions were analysed and the phylogenetic tree was generated using Geneious Prime software. Results: Fifty-eight sequences were analysed. Of these sequences, 79% (46/58) were from women who had abnormal cervical cytology. Fifteen SNPs in the LCR and eight in the E6 region were found to be the most common in all sequences. The phylogenetic analysis determined that 45% of the isolates belonged to the A1 sublineage (European variant), 34% belonged to the C1 sublineage (African 1 variant), 16% belonged to the B1 and B2 sublineage (African 2 variant), two isolates belonged to the D1–3 sublineages (Asian-American variant), and one to the North American variant. Conclusions: The African and European HPV16 variants were the most common circulating lineages in South African and Mozambican women. A high-grade squamous intraepithelial lesion (HSIL) was the most common cervical abnormality observed and linked to European and African lineages. These findings may contribute to understanding molecular HPV16 epidemiology in South Africa and Mozambique.

## 1. Introduction

It is well-established that the Human Papillomavirus (HPV) is the primary causative agent of cervical cancer [1,2]. HPV infection is one of the most common sexually transmitted infections and can naturally be cleared by the host immune system within years after acquisition [2]. The International Agency for Research on Cancer has identified more than 200 HPV genotypes and categorised them into high-risk (HrHPV) and low-risk (LrHPV) types according to their potential to induce malignancy [3]. Eighteen HrHPV types (HPV16, 18, 26, 31, 33, 35, 39, 45, 51, 52, 53, 56, 58,59, 66, 68, 73, and 83) have been identified as oncogenic and are extensively studied for their role in cervical cancer [3]. Persistent infections with these HPV types can cause a cellular change that leads to the development of cervical intraepithelial neoplasia and, eventually, invasive cervical cancer [4]. Although the importance of each genotype may differ by region, Human papillomavirus 16 and 18 (HPV16 and HPV18) are involved in more than 70% of cervical cancers worldwide, and the HPV16 genotype is one of the most prevalent in the sub-Saharan Africa region [5]. HPV16’s genome size is approximately 8000 bases in length and comprises three functional regions: a non-coding upstream regulatory region (URR), also known as the long control region (LCR) that contains regulatory elements for viral replication and transcription; an early region formed by E1, E2, and E4–E7 genes that encode core viral proteins from which E6 and E7 proteins are responsible for transformations of the host cell; a late region encoding L1 and L2 capsid proteins related to the viral DNA packaging and assembling in an icosahedral structure [4,6,7,8].

Based on sequence diversity, HPV16 has been grouped into four phylogenetic lineages, A, B, C, and D, and all of them have been implicated in cervical carcinogenesis [9]. These lineages differ between 1.0% and 10% of nucleotides at the whole-genome level [10]. When there is between 0.5% and 1.0% of the nucleotide difference between two variants from the same lineage, they become divided into sublineages [10,11]. The four identified variants of HPV16 are divided into sublineages A1–3 (formerly termed European), A4 (Asian); B (African-1), C (African-2), and D1–3 (North American, Asian-American) and have been associated with different cervical precancer and cancer risk [9]. Even within HPV16 variants, genetic polymorphisms may play a key role in infection persistence and oncogenic potential [9,12].

Geographically and ethnically, HPV16 variant distribution and associated carcinogenicity varies worldwide [9]. The European HPV16 A1, A2, and A3 sublineages account for the majority of HPV16 infections. At the same time, non-European B, C, and D variants are associated with an increased risk of cancer progression and severity of cervical lesions compared to the European variants [9,13,14]. Different nucleotide mutations in the HPV16 LCR related to altered pathways involved in viral persistence and cancer development have been reported. As an example, a Chinese study described two LCR nucleotide mutations (G7193T and G7518A) which were the potential binding sites of FOXA1 (forkhead box protein A1) and SOX9 (sex-determining region Y-box 9) transcriptional factors, respectively [15]. Furthermore, single nucleotide polymorphisms (SNPs) and nucleotide duplications in the LCR and E6 sequence regions were directly related to cervical cancer severity [16,17].

HPV16 is one of the most prevalent genotypes in South Africa and Mozambique, but there is a lack of information regarding the distribution and circulation of the HPV16 variants [18,19]. Knowledge of HPV16 variants circulating in a specific geographical region is one of the most important tools for cervical cancer prevention goals. This study aimed to investigate the HPV16 variant distribution in cervical samples collected from women with normal and abnormal cervical cytology in South Africa and Mozambique and to assess the phylogenetic relationships among variants.

## 2. Materials and Methods

### 2.1. Population Study and Samples

Samples were collected from women aged 30–98 years who were attending the community health clinic and the referral clinic within the OR Tambo district municipality, Mthatha, for cervical cancer screening or any other reason between September 2017 and August 2018. Samples were collected using cervical brushes for HPV testing and were stored in a Digene transport medium (Qiagen Inc., Gaithersburg, MD, USA). Additionally, non-pregnant women seeking care regarding gynaecological symptoms such as venereal pain, genital ulcers, and vaginal discharge or seeking family planning services were enrolled in health facilities in the Mavalane Health area in Maputo between February 2018 and July 2019. Samples were collected using cervical brushes that were stored in BD SurePath Collection Vials (Becton, Dickinson and Company, Franklin Lakes, NJ, USA). All the study samples were stored at −80 °C until HPV genotyping. Pap smear and colonoscopy exams were performed to assess the cervical cytological alterations that were classified according to the 2001 Bethesda System.

### 2.2. DNA Extraction and HPV Genotyping

DNA extraction was performed using a MagNA Pure Compact (Roche Diagnostics, Indianapolis, IN, USA) and the MagNA Pure Compact Nucleic Acid Isolation Kit (Roche Diagnostics, IN, USA) following the manufacturer’s instructions. HPV genotyping was performed using a multiplex HPV Direct Flow CHIP Kit (Vitro Master Diagnóstica, Sevilla, Spain) through the amplification of a fragment in the viral region L1 of HPV using polymerase chain reaction (PCR) according to the manufacturer’s instruction. Then, Hybridisation onto a membrane with DNA-specific probes was performed using the DNA-Flow technology for manual HybriSpot platforms according to the manufacturer’s instructions.

### 2.3. DNA Amplification and Sequencing

HPV16 positive DNA samples were subjected to amplification in a 50 μL reaction mix containing Prime Star GXL DNA Polymerase (Takara Bio, Tokyo, Japan), 0.75 μM of each primer (16-F101 and 16-R20) [20], and 5 μL of the template sample. Thermocycling conditions used were: 95 °C for 5 min followed by 40 cycles of 98 °C for 30 s, 63 °C for 30 s, 72 °C for 2 min, and final elongation at 72 °C for 5 min. Two percent agarose gel stained with ethidium bromide solution (Sigma Aldrich, Milwaukee, WI, USA) was used to visualise the amplicons. Nucleotide sequences were obtained through the Sanger sequencing method (BigDye Terminator Cycle Sequencing kit v1.1), and the quality of the resulting sequence fragment corresponding to an 1160 bp stretch covering the entire LCR and the 300 nt of the E6 ORF was analysed using FastQC software (Babraham Bioinformatics, V 0.12.0). Sequences with insufficient quality scores were discarded after their analysis repetition.

### 2.4. Phylogenetic and Statistical Analyses

The study sequences were aligned with 16 reference sequences belonging to HPV16 sublineages: KU053910, KU053914, HQ644298, and AF536180 from C lineage (African 1); AF472509, KU053922, HQ644244, and KU053921 from B lineage (African 2); AF536179, NC001526, and HQ644236 from A1–3 lineages (European); KU053933, AY686579, AF402678, and AF534061 from the A4 lineage (Asian), and HQ644257 from D lineage (Asian-American). All the reference sequences were obtained from Papilloma Episteme (PaVE, https://pave.niaid.nih.gov, accessed on 21 June 2023). Nucleotide alignment was performed using Geneious prime software (Dotmatics, V.2022.2). A Maximum Composite Likelihood phylogenetic tree was generated in MEGA 11 [21] using the UPGMA method [22]. The reliability of the observed clades was shown with internal node bootstrap values of ≥70% (after 1000 replicates). Graphics were generated using GraphPad Prism version 7.2 (GraphPad Software, San Diego, CA, USA). The prevalence of HPV16 and its lineages was calculated. Categorical variables were summarised using percentages as appropriate. When the data were presented as proportions of the total sample, the missing data were excluded from the denominator.

All sequences were submitted to the NCBI database, and their accession numbers in GenBank range from PQ178098 to PQ178155.

## 3. Results

### 3.1. Study Population

The demographical data of the South African participants can be accessed in Taku et al. 2020 [23], and for Mozambican participants in Maueia et al., 2021 [24]. Briefly, the median interquartile range (IQR) age was 46 (38–55) years for the South African and 38 (14–62) for the Mozambican study population. Of the 104 HPV16 positive samples, 58 showed acceptable DNA quality scores for inclusion (52 from South Africa and 6 from Mozambique). Of the 52 South African participants, 96% (50/52) had an abnormal cervical cytology result, with a high-grade squamous intraepithelial lesion (HSIL) (31/52. 60%) being the most common (Table 1).

### 3.2. Phylogenetic Analyses

A phylogenetic analysis of the 52 variants isolated in South Africa and 6 isolated in Mozambique aligned to reference sequences (Figure 1) showed that 50% of all the isolates (29/58) belonged to B and C lineages (African variants). Specifically, 34% of isolates (20/58) were clustered in the C1 sublineage (AF472509) (African 1 lineage). Additionally, from the B lineage, 14% of the isolates (8/58) were clustered in the B1 sublineage (AF536180), and 2% of the isolates (1/58) were clustered in the B2 sublineage (HQ644298). Notably, 45% of all the isolates (26/58) belonged to the A lineage (European variants) and were clustered in the A1 sublineage (NC-001526). Five per cent of the isolates (3/50) belonged to the D lineage (Asian-American variants), with 3% (2/58) in the D3 sublineage (AF402678) and 2% (1/58) in the D1 sublineage (HQ644257). This study found no isolates that matched the remaining sublineages under investigation.

### 3.3. Relationship between Cytology and HPV16 Lineage Distribution

Most of the study participants (55%) had high-grade squamous intraepithelial lesions (HSILs) as a cervical abnormality, and 45% had non-HSILs (Table 2). Of the 45% (26/58) of sequences that clustered to A lineage (European variants), 24% (14/58) were collected from patients with HSILs and 21% (12/58) from non-HSIL patients. Of the 50% (29/58) of sequences that clustered to the B and C lineages (African variants), 29% (17/58) were collected from patients with HSILs and 21% (12/58) from non-HSIL patients.

### 3.4. LCR and E6 Regions Nucleotide Sequences

All studied sequences were compared to the 16 reference sequences from the HPV16 sublineages to analyse the presence of single nucleotide polymorphism (SNPs) in the LCR and E6 regions. In the LCR, a total of 49 SNPs were detected (Table 3), and all sequences showed the presence of more than one SNP. The 15 most frequently detected SNP percentages and the specific sublineages are in bold in Table 3.

In the E6 region, a total of 12 SNPs were detected (Table 4), and all sequences showed the presence of more than one SNP. Eight of them were the most frequently detected. The percentages and the specific sublineages are in bold in Table 4.

### 3.5. SNPs Distribution According to the Sequence Lineages

Of the 15 most prevalent LCR SNPs, seven (G74054, G7437A, C7605, C7702T, G7750T, and C7853T) were found mostly in sequences that clustered to the A1–3 sublineages (European variants), while 8 SNPs (C7303C, A7351G, C7401A, T7585C, A7742G, C7753A, G7755A, and A7792C) were mostly found in the B and C lineages (African variants). The SNP A7792C was found strictly in the C1–4 sublineages (African 2 variants) (Appendix A. Appendix A). In the E6 region, four (G146T, T287A, A290G, and C336T) of the eight most prevalent SNPs were found in the A1–3 sublineages (European variants), and the other four (C110T, T133G, G144C, and G404A) were found in the B and C lineages (African variants). The G144C and G146T SNPs were found in the E6 PDZ domain responsible for the P53 binding.

## 4. Discussion

Globally, HPV16 is one of the most common HrHPV genotypes implicated in cervical cancer [14]. In the sub-Saharan African region, HPV16 has been the most implicated in all cervical abnormalities [27,28].

The HPV16 genotype can be divided into four main variant lineages and sixteen sublineages, differing in the whole genome sequence by less than 10% for main variants and as little as 0.5% for sublineages [29,30]. The distribution of major variants around the globe is known to follow specific geographic and ethnic distribution patterns [30,31]. The present study reports the HPV16 lineages according to the cervical cytology in a group of women from South Africa and Mozambique. Our findings showed that 79% (46/58) of the participants had any cervical abnormality, and 55% (31/58) were participants with HSILs. This high prevalence is corroborated by evidence from several HPV16 identification studies conducted in the region [5].

Clearly, the European variant is known to be spread worldwide, except for Sub-Saharan Africa, where the African variants are more prevalent, with a median of 57% of the cases. This percentage is closer to our study’s finding (50%) [32,33]. Furthermore, the European lineage (A lineage and A1–3 sublineages) has been implicated in several cases of cancer worldwide since the evolutionary genetic variation within HPV16 has already been linked by studies using partial sequencing to substantial differences in cervical carcinogenicity [31,34].

From 58 participants analysed in our study, the B and C lineages (African 1 and 2 variants) were the most found (50%), followed by the A lineage (European variant) with 45% of the cases. According to the literature, this is a normal fact since HPV16 variants show diverse geographical and ethnic distribution [30,35]. The Asian-American variant is mostly found in Central and South America, the Asian variant is principally detected in Southeast Asia, and the African in Africa and the European are the most prevalent variant in all other regions excluding Africa [16,17,35].

The sequences of the current study belonged to samples collected from a non-homogenous group. Some of the study samples were collected from women seeking STI treatment in the STI clinics and others from women seeking cervical cancer screening in community clinics and referral hospitals without real knowledge of their cervical cytological conditions. However, looking at the distribution of the lineages according to the cytology, our study results suggest that both African (C1 sublineage) and European lineages (A1 sublineage) accumulated most of the cervical abnormality cases, mostly HSILs. It was shown previously that the European variants do not follow a uniform pattern with regard to leading abnormalities, and the A4 sublineage was linked to an increased risk of cancer compared with the A1/A2 clade [35,36]. The African variants also displayed heterogeneity for disease outcomes, with the B lineage being associated with a statistically significantly reduced risk of abnormalities inducing compared with the A1/A2 sublineages [17,28]. In contrast, the C lineage conferred a statistically significant elevated risk, while the D lineages were associated with a substantially higher risk of precancer/cancer compared with the A1/A2 sublineages [34,35]. In this way, our findings may be because our study was exclusively related to HPV16 sequences.

HPV is one of the most common sexually transmitted infections in sub-Saharan Africa [5]. Despite these infections often being asymptomatic and clearing spontaneously, infections by high-risk genotypes such as HPV16 and HPV18 can progress to anogenital, oropharyngeal, and cervical cancers [37]. Furthermore, individual-level risk factors such as sexual network characteristics, the higher number of sexual partners, and sexual intercourse frequency are common in our study areas, and they can influence the spread of the infection [38,39]. Looking at the current study’s genealogical tree, we can see a closer relationship between some sequences (NM 135 and NM168; NM 085 and NM 039 in the C1 sublineage cluster; NM 080 and NM 131, NM133 and Nm 191; NM 088, NM 139 and BE 395 in the A1 sublineage cluster), suggesting that they could be related to the same sexual network.

The most frequent SNPs in our study samples were found in the LCR, specifically in the European and African lineage isolates. Additionally, those SNPs were found in samples with HSILs as cytology abnormalities (Appendix A. Appendix A: HPV16 SNPs according to the cytology). The LCR is the most variable region of the HPV genome, and it plays an important role in viral transcription and replication, as well as the persistence of the viral infection and the risk of progression to cervical cancer [40,41]. Furthermore, the LCR contains important viral transcription and replication regions, including the enhancer, E2 binding characterised by containing the ACC(N)6GGT fragments and origin of replication (ori) sites [41,42,43]. Some nucleotide changes are responsible for increased transcription activity in some HPV16 variants [36]. In our study, three E2 binding sites (fragments 7452 to 7463, 7843 to 7854, and 7859 to 7871) in the LCR were identified. However, no SNPs were identified in any of these E2 binding sites.

Different nucleotide mutations in the LCR have been reported, some of them related to altered pathways involved in viral persistence and cancer development [36,44]. This can explain the high number of LCR SNPs found in our HSIL study samples. Due to the importance of LCR in the viral life cycle and reports of mutations in some nucleotide positions of the LCR, many genetic modifications have been reported that can cause changes in the viral oncogenic potency with a potentially important role in viral pathogenicity [34].

The HPV16 genome contains two important oncogenes, E6 and E7 [11]. The loss of regulation in these two genes is the cause of intraepithelial neoplasia development [41,45]. As described previously for the LCR, our European and African lineage study samples with HSILs as a cytology abnormality showed most of the E6 gene SNPs. Moreover, as other authors have described, nucleotide modifications in this region may be related to more oncogenic variants [29,46]. However, our study did not find a significant difference in terms of the frequency of the SNPs between the LCR and E6 genes for the African and European lineages. Added to this, no significant nucleotide duplication was found in any of the studied sequences that could be linked to the alterations in the variant’s pathogeneses and probable aggressivity, which subsequently led to quick cancer development. It is important to note that two SNPs in the E6 PDZ domain were identified in our study (G144C and G146T). Numerous studies have indicated that the E6 protein has many other targets in addition to inducing p53 degradation [47]. The C-terminal PDZ-binding motif is specifically conserved among E6 proteins of HrHPVs and is essential to bind and enhance the degradation of several PDZ domain-containing proteins [47,48]. Some evidence suggests that the PDZ domain-binding motif is implicated in tumorigenesis by primary human keratinocyte transformation, hyperplasia, and carcinogenesis. Also, some of the PDZ proteins are known to have tumor-suppressor functions [49,50]. The normal amino acid combination in the E6 PDZ domains contains isoleucine and three leucine amino acids [47,48]. Nevertheless, 24 of our study sequences with G146T SNPs were found to have a tyrosine amino acid in the place of isoleucine in the E6 PDZ domain, which could suggest an inactivation of the domain without affecting the carcinogenicity effect.

Several studies mentioned some specific intratypic nucleotide polymorphisms from the sequence analysis of the E6 and E7 oncogenes among the major HPV16 variants and sub-lineages. These polymorphisms were linked to increased viral carcinogenicity [11,45,51]. For example, in the A1 and A2 sub-lineages the E6: T350G SNP; in theA4 sub-lineage the E6: T178G and E7: A647G; in the B lineage the E6: G132C, C143G, G145T, T286A, A289G, and C335T, and E7: T789C and T795G; in the C lineage the E6: T109C, G132T, C143G, G145T, T286A, A289G, C335T, and G403G, and E7: A647G, T789C and T795G; in the D lineage the E6: G145T, T286A, A289G, C335T, T350G, and A532G, and E7: T732C, T789C, and T795G [51,52]. From this group of SNPs, the E6 T350G has been found in most of the sublineages, and it was the one widely studied among the Asian and European subjects [52,53]. Looking at our study of E6 SNPs, none of the previously described were found in all of the lineages and sublineages. Thus, since our study centred on the E6 and LCR genes, extensive studies on the E1, E2, E3, E4, E5, and E7 genes could provide more details regarding the presence of these SNPs in our study group and/or African subjects.

Several limitations are taken into consideration in this present study. Firstly, in this study, samples collected in the national HPV screening and other STD programs in some sites of both countries were used, while the diversity of HPV genotypes in all countries’ provinces was not comprehensively studied. Secondly, this study is a cross-sectional design, with missing demographic information regarding the cancer status of some samples, mainly the ones collected in Mozambique; therefore, more detailed longitudinal studies are suggested at a molecular level. To add to this, the small number of samples analysed in this study is another limitation to take into consideration. Finally, although LCR and E6 genes are important regulatory sequences of the HPV genome, according to the literature, the accuracy of sublineage classification should be conducted from the whole genomes or marked regions, such as E2, for sublineage classification [41,42]. Nevertheless, the study findings are interesting and can contribute to the implementation of data regarding the molecular epidemiology of HPV16 variants in both countries.

In conclusion, the present study showed that the African and European were the most dominant HPV16 lineages. Regarding the importance of mutations in the LCR and E6 of the African and European variants in developing HSILs that cause invasive cervical cancer, women infected with these variants should be examined in future longitudinal studies to obtain further information about the oncogenic potential of these dominant variants in the study countries.

## Figures and Tables

**Figure 1 viruses-16-01314-f001:**
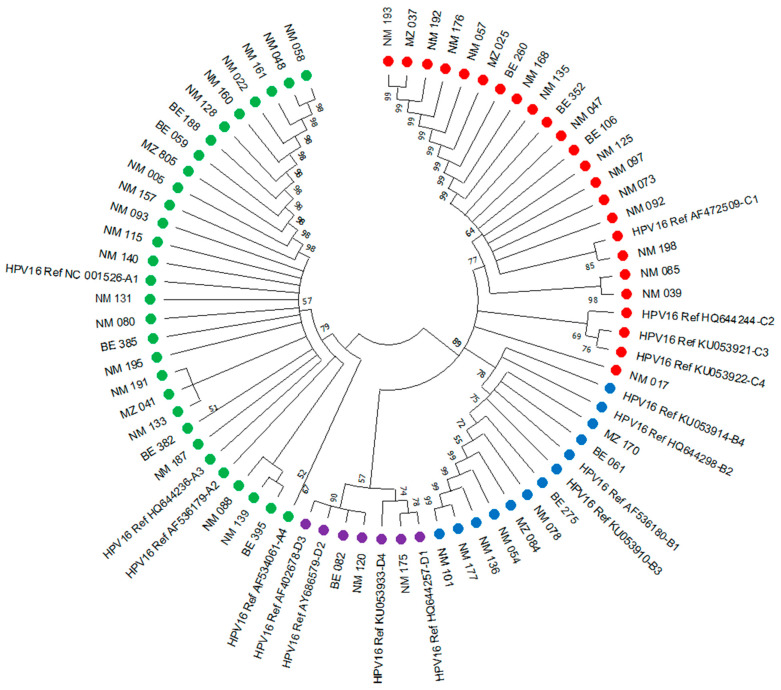
The phylogenetic relationship of HPV16 sequences isolated from 52 participants from South Africa and six from Mozambique were aligned with 16 reference sequences of HPV16. The evolutionary history was inferred using the UPGMA method [22]. The bootstrap consensus tree inferred from 1000 replicates represents the evolutionary history of the taxa analysed [25]. Branches corresponding to partitions reproduced in less than 50% of bootstrap replicates were collapsed. The evolutionary distances were computed using the Maximum Composite Likelihood method [26] and are in the units of the number of base substitutions per site. The evolutionary analyses were conducted in MEGA 11 [21]. NM represents samples collected at Nelson Mandela Referral Hospital, BE represents the samples collected at Mbekueni Community Clinic (South Africa), and MZ represents the samples collected in Maputo (Mozambique). Lineages and clusterings are indicated in coloured nodes: A: green, B: blue, C: red, D: purple.

**Table 1 viruses-16-01314-t001:** Pap smear test results of the human papillomavirus (HPV)16-positive study participants.

	HSIL *n* (%)	LSIL *n* (%)	ASCUS *n* (%)	NILM *n* (%)
South Africa (N = 52)	31 (60)	7 (13)	6 (12)	8 (15)
Mozambique (N = 6)	1 (17)	0 (0.0)	1 (17)	4 (66)

HSIL—High-grade squamous intraepithelial lesion, LSIL—low-grade squamous intraepithelial lesion, ASCUS—atypical squamous cells of undetermined significance, and NILM—negative for intraepithelial lesion or malignancy.

**Table 2 viruses-16-01314-t002:** HPV16 lineages distribution according to cytology results.

HPV16 Lineages	Cytology		
	HSIL (*n*/%)	Non-HSIL (*n*/%)	Total (%)
A1–3 (European)	14 (24)	12 (21)	26 (45)
B1–4 (African 2)	4 (7)	5(9)	9 (16)
C1–4 (African 1)	13 (22)	7 (12)	20 (34)
D1–3 (North, Asian-American)	1 (2)	2 (3)	3 (5)
Total	32 (55)	26 (45)	58

HSIL—High-grade squamous intraepithelial lesion, Non-HSIL–any cytological condition other than HSIL (LSIL—low-grade squamous intraepithelial lesion, ASCUS—atypical squamous cells of undetermined significance, and NILM—negative for intraepithelial lesion or malignancy).

**Table 3 viruses-16-01314-t003:** Nucleotide sequence variations in the LCR of the HPV16 samples (*n* = 58).

Nucleotide Position	Nucleotide Change	*n*	%	Clustering Sublineages
7179	C > A	1	1.72	
7198	G > T	2	3.45	
7209	G > T	8	13.79	
7226	G > C	3	5.17	
7226	T > C	1	1.72	
7232	C > A	2	3.45	
7252	T > A	2	3.45	
7284	C > T	1	1.72	
7288	C > A	2	3.45	
7303	A > G	2	3.45	
**7303**	**C > G**	**17**	**29.31**	**C1**
7310–7311	TT > CC	2	3.45	
**7351**	**A > G**	**19**	**32.76**	**C1**
7352	C > G	1	1.72	
7354	A > C	1	1.72	
7360–7362	TGT > GTA	1	1.72	
7366	C > T	1	1.72	
7374	T > A	1	1.72	
7388	A > T	1	1.72	
**7401**	**C > A**	**23**	**39.66**	**C1–4 and D1–4**
**7405**	**G > A**	**27**	**46.55**	**A4**
**7437**	**G > A**	**15**	**25.86**	**A1**
7552	C > A	1	1.72	
7576	G > A	2	3.45	
7582	C > A	1	1.72	
**7585**	**T > C**	**22**	**37.93**	**C1–4 and D4**
**7605**	**C > A**	**27**	**46.55**	**A1–4 and C1**
7629	G > T	1	1.72	
7630	A > T	8	13.79	
7630	G > T	1	1.72	
7645	C > A	3	5.17	
7659	G > T	2	3.45	
**7680**	**C > T**	**26**	**44.83**	**A1–4**
**7702**	**C > T**	**26**	**44.83**	**A1–4**
7715	C > G	1	1.72	
**7742**	**A > G**	**20**	**34.48**	**C1–4**
7749	G > T	4	6.90	
**7750**	**G > T**	**28**	**48.28**	**A1–4 and D2–3**
**7753**	**C > A**	**20**	**34.48**	**C1–4**
**7755**	**G > A**	**20**	**34.48**	**C1–4**
**7792**	**A > C**	**10**	**17.24**	**B1–4**
7802	G > C	2	3.45	
7813	C > A	1	1.72	
7834	C > T	3	5.17	
**7853**	**C > T**	**30**	**51.72**	**A1–4 and D1–4**
7885	G > A	1	1.72	
7895	T > C	3	5.17	
7903	G > T	1	1.72	
7904	C > T	1	1.72	

Note: % represents the proportion of the sequences where the LCR SNPs were found from the overall study sequences (N = 58). Rows highlighted in bold indicate LCR SNPs found in more than 10% of the sequences.

**Table 4 viruses-16-01314-t004:** Nucleotide sequence variations in the E6 of the HPV16 samples (*n* = 58).

Nucleotide Position	Nucleotide Change	*n*	%	Clustering Sublineages
83	C > A	8	13.79	
**110**	**C > T**	**20**	**34.48**	**C1**
132	G > A	3	5.17	
**133**	**T > G**	**20**	**34.48**	**C1**
133	C > G	8	13.79	
**144**	**G > C**	**28**	**48.28**	**B1–4 and C1–4**
**146**	**G > T**	**26**	**44.83**	**A1–4**
**287**	**T > A**	**26**	**44.83**	**A1–4**
**290**	**A > G**	**26**	**44.83**	**A1–4**
**336**	**C > T**	**26**	**44.83**	**A1–4**
351	G > T	6	10.34	
**404**	**G > A**	**21**	**36.21**	**C1**

Note: % represents the proportion of the sequences where the E6 SNPs were found from the overall study sequences (N = 58). Rows highlighted in bold indicate E6 SNPs found in more than 10% of the sequences.

## Data Availability

The original contributions presented in the study are included in the article/Appendix A; further inquiries can be directed to the corresponding author.

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
