# Peer review of "Identification of HPV16 Lineages in South African and Mozambican Women with Normal and Abnormal Cervical Cytology"

_viruses, 2024, doi:10.3390/v16081314_

Round 1

Reviewer 1 Report

Comments and Suggestions for Authors

Regarding the summary, I suggest that the abbreviations presented be written in full when they appear for the first time. For example:

  • Line 15: Human Papillomavirus type 16 (HPV16)
  • Line 17: Deoxyribonucleic Acid (DNA)
  • Line 18: Polymerase Chain Reaction (PCR)
  • Line 32: High-Grade Squamous Intraepithelial Lesion (HSIL)

I suggest that the authors do the same when abbreviations first appear in the introduction. After that, the abbreviations can be used throughout the text.

Author Response

Regarding the summary, I suggest that the abbreviations presented be written in full when they appear for the first time. For example:

  • Line 15: Human Papillomavirus type 16 (HPV16)
  • Line 17: Deoxyribonucleic Acid (DNA)
  • Line 18: Polymerase Chain Reaction (PCR)
  • Line 32: High-Grade Squamous Intraepithelial Lesion (HSIL)

I suggest that the authors do the same when abbreviations first appear in the introduction. After that, the abbreviations can be used throughout the text.

Answer: Thank you for the observation. To comply with the suggestion, the abreviations presented were wrote in full when they appear for the first time ( See the manuscript lines 15, 18, 19, and 33).

Reviewer 2 Report

Comments and Suggestions for Authors

In the manuscript entitled “Identification of HPV16 lineages in South African and Mozambican women with normal and abnormal cervical cytology the authors present significant data concerning the distribution of HPV16 variant lineages in African population. The results derived from the present analysis are interesting as little is known concerning the prevalence of specific HPV16 variant lineages and sublineages in Africa.  The article is well written and data are clearly presented. However, some points need to be addressed.

·   The authors used the Neighbor Joining method in order to construct the phylogenetic tree. In DNA sequences the Maximum Likelihood methodology is utilized with the best fit evolutionary model. The phylogenetic tree and the phylogenetic analysis must be re-evaluated.

·  The nucleotide sequences that are obtained from the present analysis are required to be submitted to NCBI GenBank sequence database in order to be publicly available. Please provide the corresponding accession numbers in the manuscript.

·   The authors described that specific HPV16 variant lineages and nucleotide variations are prevalent in the examined population. Do they find any statistically significant association between specific HPV16 variant lineages and the grade of cervical malignancy? Do specific nucleotide variations are associated with the grade of cervical dysplasia?

·     Moreover, a large scale analysis of SNPs occurring in the viral early genes, E1, E2, E4, E5, E6, and E7 and in the LCR described particular nucleotide variations that are associated with specific variant lineages, sublineages and with increased viral carcinogenicity (Expert Rev Mol Med. 2021;23:e19. doi:10.1017/erm.2021.18). Do the authors find these nucleotide changes in the examined sequences collected form African population? Please further discuss your findings considering this analysis.

Author Response

Thanks a lot for the comments. Please, find the answers in the attached cover letter

Round 2

Reviewer 2 Report

Comments and Suggestions for Authors

The manuscript has been improved by the authors and can be recommended for publication